Extensive sheep grazing is associated with trends in steppe birds in Spain: recommendations for the Common Agricultural Policy

http://orcid.org/0000-0001-6326-8942 Traba Juan 1 2
http://orcid.org/0000-0003-3247-4182 Pérez-Granados Cristian 1 3 cristian.perez@ua.es
1 Ecology, Universidad Autónoma de Madrid , Madrid, Madrid , Spain
2 Centro de Investigación en Biodiversidad y Cambio Global (CIBC-UAM), Universidad Autónoma de Madrid , Madrid, Madrid , Spain
3 Ecology, Universidad de Alicante , Alicante, Alicante , Spain
Stern David
Electronic publication date: 2022 Feb 28
Publication date: 2022
Volume: 10
Electronic Location ID: e12870
Received 2021 Oct 12; Accepted 2022 Jan 11
Copyright: © 2022 Traba and Pérez-Granados
Copyright year: 2022
Copyright holder: Traba and Pérez-Granados
License: This is an open access article distributed under the terms of the Creative Commons Attribution License, which permits unrestricted use, distribution, reproduction and adaptation in any medium and for any purpose provided that it is properly attributed. For attribution, the original author(s), title, publication source (PeerJ) and either DOI or URL of the article must be cited.
License URL: https://creativecommons.org/licenses/by/4.0/

Keywords: Chersophilus duponti, Farmland bird index, Steppe birds, Extensive grazing, Sheep, Steppes

Funding: Excellence Network Remedinal 3CM S2013/MAE2719 LIFE Ricotí ES-LIFE15/NAT/ES/000802 LIFE Connect Ricotí ES-LIFE20/NAT/ES/000133 European Commission This is a contribution to the Excellence Network Remedinal 3CM (S2013/MAE2719), supported by Comunidad de Madrid and to the LIFE Ricotí (ES-LIFE15/NAT/ES/000802) and LIFE Connect Ricotí (ES-LIFE20/NAT/ES/000133) projects, supported by the European Commission. The funders had no role in study design, data collection and analysis, decision to publish, or preparation of the manuscript.

==============================
Iberian natural steppes have traditionally been used for extensive sheep grazing, which has been noted to be positively associated with steppe bird abundance and diversity. Sheep numbers in Spain, which harbors the largest European populations of many steppe bird species, decreased by 9.2 million (37.3%) between 1992 and 2020. Steppe birds in Spain have faced dramatic declines during the same period, but there is a lack of knowledge about the potential association between sheep and open-habitat bird declines. We used sheep data from the Spanish Ministry of Agriculture and bird data (1998–2018) from the Spanish Common Bird Monitoring Program to assess the association at the Spanish scale between sheep decline and the Farmland Bird Index (FBI) and the Natural Shrub-steppe Bird Index (SBI). We also used an independent dataset on population trends of the Dupont’s Lark (Chersophilus duponti) to assess the relationship between sheep numbers and the decline of this threatened steppe specialist passerine in Spain, whose European population is restricted to Iberian natural steppes. To test for a spurious relationship between temporal series, variables were tested for cointegration. After confirming cointegration, we found a strong positive relationship between sheep abundance and the trends of the FBI and SBI indices during the period 1998–2018. The association between sheep abundance and trends of the Dupont’s Lark (2004–2015) was positive although it was not statistically significant. Although the main causes of decline of farmland and steppe birds are mainly related to agricultural intensification and land use changes, the correlation found, using two independent cointegrated datasets, between the reduction in farmland and shrub-steppe birds and sheep numbers at the country scale suggests that the decline of steppe birds in Spain may be also associated with the decline in sheep numbers. This agrees with previous studies that found a positive relationship between intermediate levels of sheep grazing and steppe bird abundance in Iberian steppes. Further research (e.g. experimental studies) is needed to corroborate our study and identify the most appropriate level of grazing intensity for protecting the most farmland and shrub-steppe birds. Our results suggest that the promotion of extensive grazing should be considered as a key factor in future Common Agricultural Policy reforms and conservation programmes to protect steppe birds.

Introduction

European natural and semi-natural open habitats, such as steppes, uplands and moorlands (natural steppes hereinafter), are generally the result of altitudinal/climatic and edaphic tree-limiting conditions, and regular low-intensity human disturbances such as low-yield agriculture, fire or livestock (Carboni et al., 2015). All these factors have shaped the characteristic open, treeless landscape of such habitats (Newton, 2004; Douglas et al., 2017), which harbors high biodiversity values (Carboni et al., 2015). In these economically disadvantaged areas, sheep systems have been decisive in maintaining rural livelihoods, but also providing several ecosystem services, such as protecting biodiversity and maintaining natural resources (Spiegel et al., 2019). However, sheep farm income in the European Union is supported by public subsidies, such as the Common Agricultural Policy (Bertolozzi, 2021) (CAP hereinafter), which makes the sector strongly dependent on changes in the policy framework (De Rancourt et al., 2006; Castillo et al., 2008; Reino et al., 2010).

Several works have suggested the existence of a complex system of multiple interactions where extensive sheep grazing affects plant structure spatial heterogeneity and plant species composition (Kurek et al., 2016; Adler, Raff & Lauenroth, 2001; Bugalho et al., 2011), as well as arthropod abundance both directly through its depositions and indirectly, by increasing habitat heterogeneity (Bugalho et al., 2011; Vickery et al., 2001; Dennis et al., 2008). Heavy levels of grazing have proven to be disadvantageous for a large number of farmland and steppe birds, due to overconsumption of plants and high fertilization (Fuller, 1996). However, light or moderate grazing may have positive effects since grazing decreases vegetation height (Vickery et al., 2001; Leal et al., 2019), and facilitates the occurrence of coprophagous species (De Rancourt et al., 2006; Jay-Robert et al., 2008; Perrin et al., 2019). Moreover, the foraging efficiency of steppe birds increases in more open habitats (Fuller, 1996; Haworth & Thompson, 1990; Buckingham & Peach, 2005; Zbyryt, Sparks & Tryjanowski, 2020) and greater visibility in sparser vegetation minimizes the risk of predation for ground-standing birds (Buckingham & Peach, 2005; Whittingham & Evans, 2004). All of these factors may favor the abundance and space use of insectivorous birds (Newton, 2004; Douglas et al., 2017; Gómez-Catasús et al., 2019; Smith et al., 2020). As a result, numerous threatened bird species have their strongholds in these extensively sheep-grazed natural steppes (Velado-Alonso et al., 2020). Of particular conservation concern are the Iberian natural steppes (shrub steppes, sensu (Traba, Sastre & Morales, 2013)), which, as with many Mediterranean calcareous grasslands, are the consequence of both anthropogenic and natural processes, with an important role played by extensive grazing by sheep in the spring and autumn (Suárez, 1994; Sainz Ollero, 2013). In Spain, sheep are traditionally reared in extensive or semi-extensive systems due to the hardiness of the autochthonous breeds and their good adaptation to adverse environmental conditions (Castillo et al., 2008).

Iberian natural steppes are among the European landscapes with the highest value for biodiversity conservation, since they harbor the main (or the entire) European population of several threatened steppe bird species, such as the Dupont’s lark (Chersophilus duponti), the Greater short-toed lark (Calandrella brachydactyla), the Stone curlew (Burhinus oedicnemus), the Little bustard (Tetrax tetrax), both the Black-bellied (Pterocles orientalis) and Pin-tailed sandgrouses (Pterocles alchata), and the Tawny pipit (Anthus campestris), among many others (Santos & Suárez, 2005). Steppe and farmland birds are the most threatened group of birds in Europe (European Environment Agency, 2015) and overall 83% of the steppe bird species show an unfavorable conservation status in Europe (Burfield, 2005). Agricultural intensification, together with habitat loss and land-use changes have partly caused the large declines detected over the past several decades for open-habitat bird populations (Carboni et al., 2015; Santos & Suárez, 2005; Fuller & Gough, 1999; Traba & Morales, 2019). In marginal regions such as those of Iberian natural steppes, where agricultural income has been traditionally low, agricultural intensification is also occurring at a field scale (Reverter et al., 2021). However, the negative trends for several steppe birds that only inhabit primary stages of vegetation succession are especially linked to shrub encroachment (Robinson & Sutherland, 2002; Herrando et al., 2014; Holmes, Maestas & Naugle, 2017). The spread of shrubs and dominant graminoids is one of the major consequences of grazing abandonment, altering local floristic and edaphic conditions (Carboni et al., 2015; Koch et al., 2015) and ultimately affecting to the entire animal community (Robinson & Sutherland, 2002).

During the second half of the 20th century, land abandonment due to rural depopulation generated a reduction in extensive sheep farming in Spain, as well an increase in stocking rates (Soto et al., 2016; Martínez-Valderrama et al., 2021). However, changes in CAP subsidies during the 21st century have also led to important structural changes in the sector (De Rancourt et al., 2006). An example is the decoupling of CAP payments in 2003, which provoked a decrease in sheep numbers in less-favoured areas, while the persistence of CAP subsidies coupled to cattle and beef-cattle production has provoked a shift from sheep-based to cattle-based systems in permanent pastures (Ramos et al., 2021; Faria & Morales, 2020; Mújica et al., 2015), usually with an increase in stocking rate. All of these changes have had a great impact on land use (Soto et al., 2016; Riedel, Casasús & Bernués, 2007) and have led to an increase in shrub and tree cover in these areas (Martínez-Valderrama et al., 2021). While a number of studies have assessed the relationship between grazing pressure and bird abundance in farmlands and uplands in the United Kingdom (Newton, 2004; Douglas et al., 2017; Fuller & Gough, 1999), our current knowledge about the association between grazing and farmland and steppe birds in Mediterranean landscapes is still limited (see Leal et al., 2019; Riedel, Casasús & Bernués, 2007; Reino et al., 2010).

In this study, we aimed to assess the variation in steppe bird abundance in relation to changes in sheep abundance in Spain over the period from 1998–2018. We estimated the nation-wide relationship between trends in sheep abundance with an annual index in population trends estimated for 35 common farmland bird species (Farmland Bird Index, Table S1) and for 20 common bird species typical of Iberian natural shrub-steppes (Steppe Bird Index, Table S2) to evaluate the potential impact of changes in sheep numbers on abundance of open-habitat bird species. Additionally, we also tested the relationship between trends in sheep numbers and population trends of the Dupont’s lark (Chersophilus duponti). We selected this species because it is a threatened habitat-specialist whose European population is restricted to Iberian natural shrub-steppes (Gómez-Catasús et al., 2018), and for which accurate data have been collected during the last 15 years (Gómez-Catasús et al., 2018). Moreover, the promotion of extensive grazing has been traditionally described as one of the main conservation interventions to maintain optimal vegetation structure for the Dupont’s lark (reviewed by Pérez-Granados & López-Iborra (2021)). Based on the assumed positive impact that low-to-moderate grazing pressure has on steppe bird abundance (Vickery et al., 2001), we expected to find a close association between the decline in sheep numbers and the decline in the population indices estimated for the community of farmland and steppe birds, as well as for the Dupont’s lark. Our results might be useful in discussing the relationship between sheep numbers and farmland and steppe birds and to formulate recommendations for agri-environment schemes for these threatened groups of species.

Material and Methods

Study area

We focused the study in Spain, since this country is host to the largest proportion of steppe birds in Europe (Santos & Suárez, 2005) and is ranked second in terms of numbers of sheep (ca. 19% of the total (Eurostat, 2020)).

Estimation of trends in sheep numbers

Annual data on total number of sheep in Spain were obtained from the General Directorate of Agricultural Productions and Markets (GDPME), of the Spanish Ministry of Agriculture, Fisheries and Food MAPA (MAPA, 2020), for the period 1992–2020. Data come from surveys collected from all of the country’s agrarian districts, and later scaled up to the province level. This is the only exhaustive nation-wide information on livestock numbers and trend. We restricted our analyses to sheep livestock and did not include goat livestock, since goat numbers are low in Spain (ca. 17% in respect to sheep livestock for the year 2020) and mainly restricted to certain regions (MAPA, 2020). We estimated the trends in sheep numbers in Spain by calculating the trend (%) since: (a) 1992, for the whole time period, (b) 1998, for comparisons with Farmland and Shrub-steppe Bird Indices (see below), and (c) since 2004 for comparisons with Dupont’s lark trends, using each mentioned year as the reference value (zero) in each analysis.

Estimation of trends in bird populations

For estimating farmland and steppe bird trends, we used bird data from the Spanish Common Breeding Bird Monitoring Program (SACRE), which is hosted by the Spanish representative of BirdLife (SEO/BirdLife). SACRE data is provided by volunteers who perform bird censuses in a set of 10 × 10 km cells distributed across the country (see Fig. S1). These sites are sampled annually, during the breeding season, following a standardized methodology. The program currently involves over 1,000 sites (Gordo, 2018). Although SACRE started in 1996, we selected data for the period 1998–2018, since during the first 2 years of the program data were collected from a reduced number of sites.

Census data are analysed by SEO/BirdLife, which provides a bird population abundance index for each species and year, estimated using the Trend and Indices for Monitoring data (TRIM) software by fitting log-linear regression models to count data with Poisson error terms (Pannekoek & Strien, 2006). These indices were converted on an annual trend (%), using the year 1998 as baseline and following the same method described to estimate trends in sheep numbers. The trend of the TRIM population index can be used as an estimate of annual variations in the abundance of bird species (Gordo, 2018; Stjernman et al., 2013) (see (Traba & Morales, 2019) for a similar procedure). Indeed, the SACRE program provides the best information of population trends for common breeding bird species in Spain (SEO/BirdLife, 2016; Casas et al., 2019). The indices of a group of species can be summarized into a single estimate to analyze trends of related bird species (Bowler et al., 2019). Among these multi-species indices, the Farmland Bird Index (FBI) is notable. This is a summary population index that includes information of species classified as common farmland birds (e.g. Stjernman et al., 2013). The FBI is an official index, adopted by the European Union, of the quality of EU’s agroecosystems for biodiversity and the effectiveness of agri-environmental interventions applied under European CAP (Eurostat, 2020). We used the official data provided by SEO/BirdLife for the FBI index in Spain, which comprised data for 35 common farmland bird species (see Table S1). Complementarily, we built a combined population index for a subset of 20 common species typical of Iberian natural shrub-steppes (Steppe Bird Index, SBI, see Table S2; for species selection see (Traba, Sastre & Morales, 2013; Santos & Suárez, 2005; Traba et al., 2007)), to further explore the relationship of birds typical of shrub-steppes with the variation in sheep numbers. The data about bird population index for each species and year was also provided by SEO/BirdLife.

Finally, we used the Dupont’s Lark (Chersophilus duponti), as a threatened habitat-specialist of Mediterranean natural steppes, to evaluate the relationship between the abundance of this strong steppe-specialist lark and changes in sheep numbers. Due the difficulty of censusing this species using traditional surveys applied during the SACRE (Pérez-Granados & López-Iborra, 2017), and thus the uncertainty in SACRE’s Dupont’s lark data, we used an independent dataset collected using a standardized, species-specific, counting method. Trend of Dupont’s Lark population was estimated by Gómez-Catasús et al. (2018) over the period 2004–2015 in Spain, using the year 2004 as baseline. This dataset comprised 42% of the Spanish populations of the species, which can be considered as representative of the population trends of the species in Spain (Gómez-Catasús et al., 2018). Annual trends were also estimated using the software TRIM (Gómez-Catasús et al., 2018), similar to the method described for the SACRE program.

Statistical analyses

The application of regression models to time series can generate spurious relationships between the considered variables, which although statistically related, may be purely coincidental (Granger & Newbold, 1974). Durbin Watson is a test statistic used to detect the presence of autocorrelation in the residuals of a regression analysis. If no causal relationship between the variables in a regression exists, the variance increases over time (non-stationarity) and affects the Durbin Watson statistic, whose low values may be due to this problem, and the coefficient of determination, which may reach high values close to 1 (Granger & Newbold, 1974). To deal with these problems, we carried out cointegration analyses between variables in the regressions described below. Two variables are cointegrated when they increase or decrease synchronously and maintain their relationship over time, suggesting a causal relationship (Engle & Granger, 1987). That is, when subtracting its expected value from the dependent variable, it is stationary (that is, mean value is stable along the time series) (Engle & Granger, 1987). We used the Johansen’s method, which assesses the validity of the cointegrating relationship using a maximum likelihood estimates approach, to determine whether both variables are cointegrated. The null hypothesis for such test is that there is no cointegration between variables (Johansen, 1988). Once it is determined if two temporal series are cointegrated, traditional regression methods can be interpreted.

We fitted a linear regression to estimate the trend of sheep numbers over the period 1992–2020. We fitted independent linear regressions to assess the relationship between bird population trends (FBI, SBI) and sheep numbers, using trends of bird population as a variable response and the trend in sheep numbers as an independent variable over the period 1998–2018, using the year 1998 as a reference value (0). Finally, we assessed the relationship between the trends for Dupont’s lark and sheep numbers through linear regression. We used the overall trends of Dupont’s lark as a response variable and the trend in sheep numbers as an independent variable over the period 2004–2015, using the year 2004 as a reference value (0).

All statistics were performed with R 3.6.2 (R Development Core Team, 2019), using the package lme4 (Bates et al., 2015) for linear regressions and packages vars and urca for cointegration analyses (Bernhard, 2008). The level of significance was p < 0.05.

Results

Sheep numbers significantly declined in Spain during the period 1992–2020 (Fig. 1). In 2020, there were 37.3% fewer sheep than in 1992 (−9,175,782 sheep; linear regression, F-statistic = 128.2.10, df = 27, adjusted R2 = 0.82; p < 0.001, see Table S3 for full statistics, Fig. 1). The Johansen test showed significant cointegration between bird indices and sheep trends, which indicates that our results are unlikely to be due to spurious relationships: Spanish farmland bird trend (FBI) vs. Sheep trends Johansen test = 26.82; p < 0.01; Spanish steppe bird trend (SBI) vs. Sheep trend: Johansen test = 27.31; p < 0.01; Dupont’s lark trend (FBI) vs. Sheep trend: Johansen test = 25.02; p < 0.01.

Figure 1 Temporal trend in the number of sheep in Spain during the period 1992–2020.

The linear regression is shown in blue, and 95% Confidence Intervals in grey (linear regression: adjusted R2 = 0.82; p < 0.0001).

We found a strong, positive and significant relationship between the trend of Spanish farmland birds (FBI) during the period 1998–2018 and the sheep trend (linear regression, F-statistic = 39.85, df = 19, adjusted R2 = 0.66, p < 0.001, Fig. 2A, Table S3). Similarly, a positive and significant association was found between the trend of natural steppe birds (SBI) and the sheep trend for the same period (linear regression, F-statistic = 27.4, df = 19, adjusted R2 = 0.57, p < 0.001, Fig. 2B, Table S3). The linear relationship between Dupont’s lark trend and the sheep trend over the period 2004–2015 showed a positive relationship although it was not statistically significant (linear regression, F-statistic = 4.44, df = 9, adjusted R2 = 0.26, p = 0.06, Fig. 3, Table S3).

Figure 2 Linear relationship between trends of bird communities and trend of sheep abundance in Spain during the period 1998–2018.

All variables are represented as the percent reduction using the year 1998 as the reference value (0). (A) Farmland Bird Index (adjusted R2 = 0.66; p < 0.0001). (B) Steppe Bird Index (adjusted R2 = 0.56; p < 0.001).

Figure 3 Linear relationship between Dupont’s Lark trend and trend of sheep abundance in Spain during the period 2004–2015 (adjusted R2 = 0.25; p < 0.0649).

Both variables are represented as the percent reduction using the year 2004 as the reference value.

Discussion

Our results show a strong decline in sheep numbers in Spain during the last three decades, which is in agreement with the decreasing number of sheep livestock in many European countries: 15% for the period 2002–2018 (Eurostat, 2020; Pollock et al., 2013; Varga & Molnár, 2014; Vargaa et al., 2016; De Arriba & Barac, 2018). Nonetheless, the sheep decline detected in Spain has been especially steep since 2007, firstly, and even more so since 2009, when the number of sheep drastically declined (Fig. 1). This decrease in sheep numbers may be related to changes in CAP subsidies and largely explained by the uncoupling of sheep subsidies that started in 2006 and become permanent in 2010 (Mújica et al., 2015). After controlling for spurious correlations, we found a very strong and positive relationship between the reduction in sheep numbers and the decline in farmland and shrub-steppe birds at the national scale. The relationship between sheep numbers and the population trend of the Dupont’s Lark, a strong habitat specialist of Mediterranean natural steppes (Suárez, 2010), was just close to significant (p = 0.06). This result was obtained using an independent dataset (Gómez-Catasús et al., 2018) and at a shorter temporal scale (11 vs. 20 years) than FBI and SBI analyses. The lack of a significant relationship for the Dupont’s Lark case might be partly related to the more restricted distribution range of the species, which may make it difficult to detect nation-wide changes. Likewise, the decline in the Dupont’s Lark has been associated with other land-use changes due to human activity, such as habitat loss due to ploughing (Tella et al., 2005) and the development of wind farms (Gómez-Catasús, Garza & Traba, 2018), which may have muddled our results.

We are aware that the decline in farmland and shrub-steppe birds in Spain is not exclusively explained by the decline in sheep numbers, but our results and previous studies strongly suggest that the significant relationships found are not simple correlations. It is well known that the negative trends in farmland and steppe birds in Europe are mainly related to agricultural intensification and land use changes, such as the decline in fallow lands (Traba & Morales, 2019; McMahon et al., 2010), which traditionally were an important grazing source (Ramos, Robles & González-Rebollar, 2010), as well as the increase in the use of pesticides and herbicides (Boatman et al., 2004; Chiron et al., 2014), higher mechanization (Santangeli, Di Minin & Arroyo, 2014), or the increase in trellis vineyards and irrigated woody crops, which are largely unsuitable habitats for the considered groups of birds (De Frutos, Olea & Mateo-Tomás, 2015; Casas et al., 2020), to cite a few. Velado-Alonso et al. (2020) recently found strong geographical associations between steppe bird richness and local sheep breed richness, which may be interpreted as an indicator of the intensity of extensive sheep grazing. Though these relationships are probably mediated by other environmental gradients, sheep grazing could have effects on increasing habitat heterogeneity, which could help to promote steppe bird richness and abundance (Velado-Alonso et al., 2020).

The associations found between the decline in both sheep numbers and farmland and shrub-steppe birds in Spain are in agreement with the well-recognized positive impact that extensive grazing has on the distribution and abundance of farmland and steppe birds (Douglas et al., 2017; Fuller, 1996; Haworth & Thompson, 1990; Traba, Sastre & Morales, 2013; Santangeli et al., 2019) (see Fuller, 1996; Evans et al., 2006) for a negative effect of both heavy or very low levels of grazing in steppe birds). For example, steppe birds are adapted to open habitats, which are partly maintained by light or moderate grazing (Fuller, 1996). Grazing plays an important role by decreasing vegetation height and increasing spatial heterogeneity (Bugalho et al., 2011; Leal et al., 2019; Evans et al., 2006), which may increase the foraging efficiency of steppe birds by facilitating foraging in more open habitats (Fuller, 1996; Leal et al., 2019; Zbyryt, Sparks & Tryjanowski, 2020; Murray et al., 2016) and minimizing their predation risk while foraging on the ground (Buckingham & Peach, 2005; Whittingham & Evans, 2004). Sheep grazing has been described as a main driver of these habitats, as it may favor open, creeping plant phenotypes, which disperse their seeds through the livestock dung (Malo & Suárez, 1995; González-Rebollar & Ruiz-Mirazo, 2013). The amount of plants consumed by sheep in Mediterranean shrub steppes under extensive grazing has been estimated at around 1,500 kg/ha/year (dry weight), while dung production is around 600 kg/ha/year (Le Hourèou, 1991). Moreover, sheep dung is twice as attractive to Mediterranean dung beetles as cattle dung, three times more than red deer pellets, and four times more than horse dung (Dormont et al., 2007). Thus, extensive sheep grazing will reduce plant structure and density (Adler, Raff & Lauenroth, 2001) and facilitate the occurrence of dung-processing arthropods (Prather & Kaspari, 2019), an important food source for some farmland and steppe birds (Vickery et al., 2001; Dennis et al., 2008; Jay-Robert et al., 2008). Finally, these areas might be preferred by birds feeding on insects and especially on dung beetles and other coprophagous arthropods (Leal et al., 2019; Jay-Robert et al., 2008; Gómez-Catasús et al., 2019; Smith et al., 2020).

Goat abundance was around six times lower than sheep abundance in Spain in 2018, and especially linked to specific regions. Thus, in this work we only considered trends in sheep numbers. Nonetheless, the negative relationship between Iberian steppe birds and sheep numbers was maintained when the trends in goat numbers in Spain were included, which was around −10% during the period 2002–2018 according to official data (MAPA, 2020). Besides sheep number decline, during the last decades there has also been a great change in livestock husbandry towards higher intensification (Riedel, Casasús & Bernués, 2007; Bernhard, 2008) that may alter the role that sheep play in natural steppes and other plant communities. For example, in recent times sheep are more often kept indoors than free-ranging (Martínez-Valderrama et al., 2021) and the proportion of pregnant and lactating ewes, which are usually kept indoors, has increased (Martínez-Valderrama et al., 2021; Riedel, Casasús & Bernués, 2007; Naylor et al., 2005). These changes in sheep husbandry imply a substantial reduction in the workload and may maximize farmers’ income, but reduce the intensity and duration of outdoor grazing (Soto et al., 2016; Mújica et al., 2015).

Under these circumstances, extensive sheep grazing in Spain has practically disappeared (Martínez-Valderrama et al., 2021), and the role of sheep as an ecosystem engineer species has disappeared (Naylor et al., 2005) or has even become negative due to pollution or overgrazing and soil degradation around intensive farms (O’Brien et al., 2016; Pulido et al., 2018). In Spain, shrub- and tree-encroachment has steeply increased in areas considered as transitional from former extensive grazing to current land abandonment (Martínez-Valderrama et al., 2021), and the consequences of this functional change are yet to be fully determined.

Our results suggest that the alarming decline in Iberian farmland and shrub-steppe birds might be, at least partly, associated with the sharp decline in sheep livestock occurring in Iberia. Further research is needed to understand the role of sheep grazing in steppe and farmland bird declines. In the meantime, there is a need to reverse the negative effect that this trend in sheep numbers may have on habitat heterogeneity and quality (i.e. food resources) for steppe birds. Our findings suggest that extensive grazing should be considered as a key factor in future Common Agricultural Policy reforms and conservation programs to stabilize, and even better increase, the traditional sheep farming system. Such changes in CAP policies to promote sheep farming may partially tackle other concerns in the EU, such as the preservation of biodiversity, reduction of risks due to wildfire, valorization of environmentally friendly agricultural practices and prevention of desertification (Mújica et al., 2015). Besides, changes from sheep to cattle, which could be seen as an opportunity for farmers due to lower production and maintenance costs, has to be deeply evaluated, as may provoke relevant changes in ecosystem structure and functionality (Ramos et al., 2021; Faria & Morales, 2020; Mújica et al., 2015). As the same grazing pressure may be favorable for some species of conservation concern but detrimental to others (Leal et al., 2019; Reino et al., 2010), we encourage researchers to estimate the adequate grazing intensity for protecting most farmland and steppe birds, as well as to perform species-specific studies for proposing a concrete grazing intensity to protect the most threatened species. These studies would provide scientific evidence to managers and therefore increase the implementation of extensive grazing as a conservation intervention (Gibbons, Wilson & Green, 2011). Extensive sheep grazing should be promoted as a multirole and low impact practice, which may contribute to increasing habitat heterogeneity, reversing shrub encroachment and improving the situation for birds while avoiding the need to apply other resource-consuming, and potentially hazardous practices such as mowing, manual shrub-clearing and/or controlled burning (Carboni et al., 2015). In a climate change scenario, natural steppe habitats may need habitat management actions aimed at improving habitat quality for open-habitat bird species (Leal et al., 2019; Gibbons, Wilson & Green, 2011; Menz, Brotons & Arlettaz, 2009; Pérez-Granados, Serrano-Davies & Noguerales, 2018; Hellicar & Kirschel, 2020).

Supplemental Information

Supplemental Information 1 Raw data used for analyses and figures.

Click here for additional data file.

Supplemental Information 2 Supplemental Material.

Click here for additional data file.

We wish to thank SEO/Birdlife and specifically Juan Carlos del Moral and Virginia Escandell for providing SACRE data. We are grateful to Piotr Tryjanowski, Francisco Moreira, Rita Ramos and one anonymous reviewer whose comments helped to improve the manuscript.

Additional Information and Declarations

Competing Interests

Author Contributions

Data Availability

The authors declare that they have no competing interests.

Juan Traba conceived and designed the experiments, performed the experiments, analyzed the data, prepared figures and/or tables, authored or reviewed drafts of the paper, and approved the final draft.

Cristian Pérez-Granados performed the experiments, analyzed the data, prepared figures and/or tables, authored or reviewed drafts of the paper, and approved the final draft.

The following information was supplied regarding data availability:

The raw data is available in the Supplemental File.

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
