# Peer review of "Extensive sheep grazing is associated with trends in steppe birds in Spain: recommendations for the Common Agricultural Policy"

_PeerJ, doi:10.7717/peerj.12870_

## Round 0.1 · original submission · Major Revisions

I am in the odd situation of one referee recommending I reject the paper and the other three recommending minor revisions only. Given this, I have decided on major revisions. Your revised manuscript needs to be accompanied by a detailed explanation of how you have addressed all the referee comments, but in particular, how you have addressed the critique of Reviewer 2.

·

Basic reporting

I generally like the manuscript, probably because I love both - sheep and birds and try to find interactions between both group of organisms

Experimental design

I have only one general comment - the manuscript based in fact mainly on correlation

Validity of the findings

no comment - I add directly on the MS

Additional comments

I add directly on the MS

·

Basic reporting

The manuscript would benefit from some language improvement. Also, the structure/scope of the introduction could be better organized,as you jump between topics in a not very organized manner. For example, the details on the overall impacts of grazing on biodiversity might not be needed here… But it depends on the major issue of clarifying what is the scope of the paper.

Experimental design

There is a major issue with this paper: Following the abstract and introduction, it is supposed to address issues related to birds and sheep in Iberian natural steppes, but the title and data analysis focus on country-level data on livestock and bird trends. This makes the paper confusing and not adequate for publishing as it is.

Validity of the findings

The main issue is that authors use a dataset of steppe bird abundance and sheep numbers that relates to national-level data. In that perspective, both total sheep numbers and overall bird abundance (many species are not exclusive of shrub steppes) may not represent what happens in natural steppes. This relates to the key issue; what is the geographic scope of these natural steppes? Both sheep and bird data should focus on these areas.

Additional comments

If analysis are to be carried out at national level, and if sheep numbers are an indicator of the amount of available habitat, why authors don’t focus on the latter, rather than the former ? Surely, the solution to the bird decline is not to increase the number of sheep, but the area of pastures/fallow. If sheep were replaced by cattle, would that be a problem ?

Reviewer 3 ·

Basic reporting

The research question is clear, the data and the methods are suitable for addressing the question, the results are not particularly open to interpretation, and the discussion and conclusions are well supported by the results. Well done!

I have checked the data, and I can only say that it seems to be available and correspond to the paper. Did the authors use the ‘FBI_Index’ or the ‘FBI_Index_Annual’ for their analyses? Please, clarify. Likewise for the variables ‘SBI_Index’ and ‘SBI_Index_Annual’ and for the ‘sheep-dupontvF’ sheet – which variables were used in the analyses? Please, clarify to ensure reproducibility.

Figures: Presumably, the baseline year (corresponding to annual change = 0) is the first year of each period mentioned in the figure captions, right? For example, in Fig. 2 the baseline year is 1998, right? It might be worth clarifying it in the figure captions and perhaps mentioning that the value of zero represents the baseline against which the change is estimated.

This is a bit of a cliché, but I think the writing needs a bit more work. I am not a native English speaker, but I found quite a few typos and some sentences that don’t seem to make much sense. I have included some suggestions and comments.

Experimental design

Some clarifications and further explanations on the statistical methods used are needed. In particular, what is Johansen’s method? (lines 176-177) How am I supposed to interpret the results of the Johansen test presented in lines 195-198? There are statistically significant p-values there; what do those values mean? That the variables are cointegrated? It is confusing because, in many applications, a significant p-value could imply the opposite. Please clarify since this was the method used to test for cointegration, which is essential for the paper. Likewise, what is the Durbin-Watson statistic (line 170)? Talking about statistics, please provide the point estimates, standard error, and 95% confidence intervals of all the parameters of your linear regressions. Uncertainty measures and point estimates are fundamental, and at the moment, the authors only report on the F-statistic. The authors could consider a table detailing these values.

The authors mention that the positive effects of sheep grazing on steppe birds are higher at intermediate sheep densities. Have you considered fitting models to accommodate this effect? It is relatively straightforward to by including a square sheep change rate. It would be something along the lines of SBI ~ sheep abundance + (sheep abundance)^2. This could also be done using the raw sheep abundance presented by the authors in their dataset. I reckon the authors should consider it as it could make their results and conclusions stronger.

Validity of the findings

Although the authors have done a nice job explaining the potential mechanisms underpinning their results, I think that the direct conclusions and implications are less clear. Therefore, I consider that it would be good to provide some explicit and straightforward assessment of the role of sheep in determining steppe bird declines. Do the authors think that increasing the number of sheep will result in a parallel increase in steppe birds? Are the authors recommending investing in maintaining sheep in Spain as a conservation measure? These will imply a direct role of sheep on the management and decline of steppe birds. Or can the ecosystem services provided by sheep be substituted by other conservation interventions? For example, how about regular burning? Or perhaps fostering the recovery of native herbivores such as the rabbit – would that replicate the effect of sheep on the ecosystem and help recover steppe birds?

Additional comments

I am nit-picking, but how come “natural” steppes are so reliant on human activity? I found it a bit contradictory, to be honest. Would some of those “natural” steppes revert to forest or scrubland without human use? In that case, would those be actual “natural” steppes?

Title: I think it should be “in Spain in the PAC context” The authors could add EU PAC to indicate that they are referring to the EU programme. Moreover, I suggest not using acronyms in the title since it can be confusing for the readers not familiar with the meaning of PAC.
Abstract, lines 23-24, and throughout the manuscript: I am pretty sure it should be “sheep numbers” (plural) instead of “sheep number”.
Lines 56-57: I am not sure what the authors mean by “both directly through its depositions and indirectly”. Is it that sheep dung affects it directly, and there are other indirect effects? By the way, what are those indirect effects?
Line 60: Coprophagous species? Beetles?
Line 68: the main populations within Iberia or a different geographical area? Please, clarify.
Line 86: Change “supposed” to “caused” or “produced”
Line 90: I suggest changing “reduced” to “limited”
Line 96: I suggest changing “with” to “on” – “sheep numbers on the abundance”
Line 100: I suggest changing “compiled” to “collected”
Line 118: I don’t think “husbandry” is the right word here.
Line 119: Change “reduced” to “less”, although I think this sentence could be re-written to improve clarity.
Line 124: Change “cero” to “zero”
Line 128: Perhaps change “partnership” to either “partner” or “representative”
Line 130: Change to “a set of”
Line 181: I suggest reordering the words so it reads “response variable”
Line 192: I suggest changing to “In 2020, there were 37.3% fewer sheep than in 1992”
Line 226: I don’t understand this sentence very well. What are casuistic correlations?
Line 231: I think it should be “to cite a few”
Line 246: I suggest changing “plant consume” to “plants consumed”
Line 262: I don’t understand what the authors mean by “which are usually hosted”. Please, clarify.
Line 265: I don’t understand what the authors are discussing here. Please, re-write the sentence to make it clear.
Line 275: I think the authors mean “this study”

·

Basic reporting

The paper is structurally sound and of good ecological, conservation and management importance. It uses a national dataset which gives a general overview of the tendencies in the country, strengthened the conclusions.
I definitely thing is a publishable paper although I have some concerns about the lack of support and detailed information of some decisions and thing some conclusions are somehow pushed beyond what the paper results tell us.
I also thing that, including 79 citations in a paper addressing a simple concept is too much. 40-45 citation would be reasonable and will give sufficient background and context.
Lastly, please check again the common and Scientific names of the species in Tables S1 and S2 since some names do not match (e.g., S2: Crested Lark, designed as Sylvia conspicillata), and check the trend in Spain so is the same in the two tables. Additionally, the Common Linnet as changed genus and is now Linaria cannabina, however, this change is not necessary if you add a note to the table referring you are using the older/traditional classification.

Experimental design

The research questions are well presented and explained.
The method section is clearly explained. Both sheep and bird data were collected at a country level, with the best available information considering the spatial and temporal scale of the work.
The use of an independent dataset for the Dupont’s Lark shows that the authors considered the under representation of the SACRE Dupont’s lark data and worked around it to get a sound dataset for this species.

Validity of the findings

This paper puts in highlight what has been discussed in the scientific community for a few years now, which is the decrease of both the traditional sheep extensive grazing and steppe and farmland birds.
Respecting to conclusions, the authors focus mainly on the sheep-steppe birds relationship, although they have also evaluated the relationship for farmland birds (which are not all steppe birds). More attention should be given to farmland birds in the discussion and conclusion.
In addition to that, the strong direct association of cause-effect made between both sheep and bird declines need to be revised. As the authors also mention, there are other factors that can contribute to the bird declines and that were not measured in this work (lines 224-226). Therefore, although the cause-effect is unquestionable, it should be presented as one of many other factors already known that contribute for steppe and farmland birds declines.
Additionally, I have a particular concern with the fact that the authors do not give enough attention to the shift from sheep-based to cattle-based systems occurring in the Iberian Peninsula, which is already known to negatively affect farmland bird densities and distribution (Reino et al. 2010) and is associated with the CAP subsides that the author briefly mention.

Additional comments

The lines numbers are made considering the PDF version provided

1. Title: I'm assuming PAC mean the same as CAP (Common Agricultural Policy) in which case I suggested changing to the English abbreviation.
Additionally, the authors recognise, in the discussion, that more research is needed to better understand the association of bird declines and sheep declines. They also mention the existence of other factors that lead to the observed bird decline (Lines 224-226). In this sense, I think the title should be rethought in order to not be misleading of the paper main conclusions.
On a second note, if the authors want to focus on CAP (which I agree they should), some more context of it is needed, particularly in the introduction; and further discussion between CAP and the paper main results, should be added.

2. Keywords: not sure why the authors included the “paramo” as keyword. Is not used as a term in any part of the paper.

3. Lines 76-82: is not clear to me how agricultural intensification leads to shrubs encroachment. That may be true for agricultural and grazing abandonment (as stated in lines 80-82), which tend to happen in the marginal regions mentioned, due to the low income but not do to agricultural intensification.

4. Lines 87-90: There are some recent works addressing the relation between grazing and farmland and steppe birds in Mediterranean landscapes missing throughout the manuscript. Please see papers from Faria et al., Reino et al., Beja et al. and Moreira et al.).

5. Lines 94-95: The term “common natural steppe bird species” can be misleading, since some of the species in the index are not considered steppe birds, although they can occur and are typical of Iberian natural steppes. Please use the term used in the methods “common species typical of Iberian natural steppes”.

6. Lines 150-152: The SBI has an index to measure the relation between sheep numbers and birds, that occur in natural steppe areas, is correct. However, using this index to “further explore the relationship of steppe-specialist birds with the variation in sheep number” in not correct and should not be done. First, there are some steppe-specialist birds missing from index (eg., Great bustard, Otis tarda). Second, at least half of the species used in the index are not classified has steppe birds and most are common in a diverse range of habitats (Little Owl, European Stonechat, Common Linnet). That said, using this species in the index will not allow to draw conclusion about the steppe-specialist birds. I don’t think you need to change the index, just rephrase the sentence and change the conclusions to account for this.

7. Line 256-258: If the relationship including both sheep and goat was evaluated, why not present those results? If the number of goats is “much reduced” and was not evaluated because of that (lines 117-120), why discussing those results at all?

8. Lines 277-285: The discussion and conclusion of the subject addressed in this line, although it may be true, goes beyond this study scope and findings. The authors did not measure in what extent the different types of grazing affected birds’ trends neither they worked with grazing intensity.

---

## Round 0.2 · Minor Revisions

As usual, please provide detailed responses to the referee comments. These will be especially important as I probably will only review the revised paper myself.

·

Basic reporting

I already commented it it the first turn. However now I see really improving version, although from the beginning I was quite optimistic.

Experimental design

OK, it looks like correlate study, but anyway, quite interesting. The authors underlined potential limits.

Validity of the findings

In local, as well as European scale, we need similar results to better protection of farmland birds.

Additional comments

none

Reviewer 3 ·

Basic reporting

No comment

Experimental design

Please explain in the main text what is the Durbin-Watson statistic and how to interpret it. As it is written currently (lines 207), it is impossible to know what this means or what does it imply. The authors have provided additional details in their responses letter, but they should add further explanations in the main text for the sake of interpretability and repeatability.

Validity of the findings

I have some minor comments that the authors might want to consider. These are suggestions for clarifying the interpretation and meaning of some of the statistics presented in the paper.

First, I don’t think the analysis can entirely rule out spurious correlations. Cointegration is helpful but not perfect, and as the paper is currently written, it gives the impression sometimes that the results are causal. I think the authors should be more careful and use expressions such as “to mitigate the impacts of potentially spurious relationships” or “to test for the potential effects of spurious relationships” (e.g., abstract) and “indicates that our results are unlikely to be due to spurious relationships” (line 235 in results).

I might be nitpicking, but I really dislike the concept of “marginally significant” (e.g., abstract and line 246) to refer to a p-value = 0.06. In my opinion, p-values are either significant or not, even more so when the authors themselves indicate that they are using the p < 0.05 cut off (line 228). Therefore, I suggest deleting instances of “marginally significant” throughout the manuscript, perhaps changing to “trend negatively although it was not statistically significant” or something similar.

Additional comments

Line 235: “posterior regression results” - posterior usually applies or refers to the results of a Bayesian analysis. Did the authors do a Bayesian analysis? If not, please delete the word “posterior” as it is confusing.

I have reviewed this revised version of the manuscript, and I think the authors have done a good job addressing my comments. In general, the paper is interesting, and the results are properly contextualised. I am looking forward to seeing it published in PeerJ.

·

Basic reporting

In Table S2, the Spectabled Warbler, has a typo, it should be Spectacled Warbler.

Although there was a big language improvement, some sentences are still hard to read and too long. The paper will benefit from a second revision focusing on the sentence length and structure.

Experimental design

Nothing to add

Validity of the findings

Nothing to add

Additional comments

1. The CAP section of the manuscript, particularly in the discussion, was greatly improved and know gives a good idea about that problem while making suggestions about how to improve it in the future.

2. L. 185-190 : Since the authors selected species for the SBI index taking into consideration those species that can be found in “uplands” in UK, this need to be justified in the methods. There is a lot of research on this topic in the UK, which makes it appealing to compare to, however, the Iberian Peninsula has a particular type of natural and semi-natural open habitats which are not found in the UK (steppes). Considering that and the fact that important steppe bird specialists are missing from the index (like the Great Bustard) I wonder if the authors should re-think their position about what species to include or better justify the species included in the index.
In addition, the authors maintained a misleading sentence from the previous version: “to further explore the relationship of steppe-specialist birds with the variation in sheep numbers.” As they said in the response to my previous comment the 20 birds selected are “typical of shrub-steppes” and are not all steppe-specialist birds”, so this term should be changed.

3. L. 300-302: regarding the goat results. Since the authors mention in the methods (L. 155-157) that “did not include goat livestock, since goat numbers are low in Spain” the reader will not “ask themselves if the same pattern would have been obtained if including goats in the analyses”. Moreover, including these trends in the discussion is questionable, if the numbers of goats are low, then the relationship found is because of the sheep and not the goats. Secondly, since goats are “especially linked to specific regions" should be consider since the paper looks at trends at the country level.
Considering all the above, I think the lines 298-302 should be removed from the discussion.

---

## Round 0.3 · Minor Revisions

I intend to accept the paper but think that the writing about the statistics still needs a few changes as follows:

1."To test for the potential effects of spurious relationships " Please change this to "To test for a spurious relationship" or something similar. The cointegration test doesn't test *the effects* of a spurious relationship but whether the relationship could be spurious.

2. Please change the terminology "sheep annual change rate" and similar expressions for other variables. Your variable is simply the percent change since the base year. It’s not the change year to year. You would get the same basic results using the original variables and be able to use one more observation… Things would be simpler and clearer if you just used the original variables, I’m not going to force you to do this, but you need to come up with a better description. Probably you should just put “sheep” on the graphs where you now have “sheep annual change rate” and explain that the variable is presented as the percent reduction since 1998 in the text.

---

## Round 0.4 · Minor Revisions

I'm sorry to be pedantic about this. "Change rate" is not good English. "Rate of change" implies something like a derivative such as an annual percentage change. On one axis of the graph you have "Sheep abundance" that is a good term. Please change the manuscript to eliminate the "rate of change" concept and use uniform terminology for the different variables.

---

## Round 0.5 · accepted · Accept

Thank you for making the final changes, which will make it easier for readers to understand your paper.